

# Sustainable development of multi-media communication path of broadcasting and hosting with a dynamic environment

Qiong Hu

College of Art and Design, Xi'an Mingde Institute Of Technology, Xi'an, China

## ABSTRACT

To enrich people's lifestyles at home, the research on the transmission path of new media broadcasting and hosting programs has become a hot topic. The traditional statistical regression model has low prediction accuracy and weak generalization ability on such issues. Therefore, we propose an improved comprehensive path planning algorithm based on an ant colony algorithm to search for the optimal path of the multi-media transmission for broadcasting and hosting programs in a dynamic environment. Firstly, we improve the bidirectional search strategy, optimize the probability transition and extend the early search scope. Then, we utilize the allocation strategy of the wolf to change the updating rules of the pheromone. Finally, we take the shortest process time for listeners to obtain the broadcast program as the optimization goal and construct a comprehensive evaluation model. We also solve the optimal parameters to improve the overall performance of our method for finding out the excellent path of multi-media transmission in a dynamic environment. Experiment results show that our method can achieve the optimal route plans and we can demonstrate that the path planned by the improved ant colony algorithm is more reasonable, which can effectively avoid the optimum local problem and shorten the solution.

# INTRODUCTION

Artificial intelligence technology can promote the media layout and path optimization of broadcasting and hosting programs, which is the key to promoting the sustainable development of the broadcasting and hosting industry. Multiple media modes of information dissemination have been widely adopted to enrich people's entertainment activities. Among them, broadcasting and hosting programs, as a program with the most extended existence and the most comprehensive dissemination range, need to keep up with the development of science and technology and keep up with the trend of the time. The single problem of the classic broadcast program has become increasingly prominent. Therefore, the diversification of broadcast program communication methods has become inevitable (*Arnaud, 1980*; *Srivastava & Jain, 1997*). The communication path of customized broadcasting and hosting programs aim to enrich the content of people's daily life and to close the relationship between the programs of broadcasting and to host for the people.

Corresponding author
Qiong Hu, huqiong2023@126.com

Planning routes for broadcasting and hosting programs and allowing listeners to receive them faster can make up for the shortcomings of traditional broadcasting and hosting programs in flexibility and convenience. In the early period, research on radio and host programs' communication paths focused on the operation mode and broadcast content. As an essential link and core issue of operation development, the line planning of broadcasting and hosting programs play a critical role in improving the practical value of broadcasting and hosting programs (*Freeman, Hess & Waller, 2018*; *Varlamov, 2018*).

There are two directions in the research about the transmission path of broadcast and host programs. One is to research the location and layout of communication media. *Yuning (2019)* abstracts the location layout as a clustering problem, which designs the transmission path with the principle of the lowest operating cost by the media distribution applying the K-means algorithm. In *Yuekun & Xiuyan (2020)*, they consider the cost factor account and plan the appreciative route based on the existing communication media, which can increase the utilization rate for each medium. *Yeqian, Wenquan & Feng (2012)* establishes a variable line-type communication model and use solves the model with the genetic algorithm for the operation cost and audience affection. In *Lang & Changxi (2018)*, they propose a multi-objective robust optimization model and solve the algorithm by the improved NSGA-II with a maximum number of people constraint. Note that they design their method with a severe condition: keeping the lower operating cost and more listening time of listeners. Another one is to solve the optimal routes in planning. Traditional path-planning algorithms include the Dijkstra algorithm (*Yunlong & Ying, 2009*), the algorithm in *Dieter et al. (2000)* and so on. With the development of intelligent bionics, intelligent algorithms such as the ant colony algorithm (*Zhiyong, 2016*) and particle swarm algorithm (*Yong, 2018*) have been applied to path planning. Ant colony algorithm is first employed to solve the traveling salesman problem. Ants secrete pheromones during foraging and communicate with each other through these pheromones to gradually find the optimal route using swarm intelligence. However, when the differences between pheromones in different paths expand, it is easy to be bound by the local optimal solution and the huge randomness in the early stage, resulting in slow convergence and other problems. In *Zhizhong (2018)*, they propose the idea of bidirectional search to enhance the global search ability of the model. In *Chunyang, Ping & Genrong (2020)*, the authors propose a strategy of mutation factor and an allocation strategy for the wolf group, which can improve the transfer formula and optimize the efficiency of an ant colony. In *Chaofan, Huadong & Nan (2020)*, they change the conventional mode of setting parameters with the experience and apply the particle swarm optimization algorithm to update the parameters, which can save time and improve the optimization results.

In conclusion, route optimization problems concentrate on establishing media layout and route optimization models for broadcasting and hosting programs. Unfortunately, there are few types of research on the algorithms for solving and optimizing steps for broadcasting and hosting programs. In the path planning algorithm, the ant colony algorithm can achieve high accuracy, fast convergence speed and strong optimization ability, which can solve the problem of transmission path planning for broadcasting and hosting programs well. But it has shortcomings such as trapping into local solutions quickly,

slow convergence speed and the parameter setting, which can directly affect the optimal solution. Therefore, we propose a comprehensive path-planning scheme by the improved ant colony algorithm. We enhance the global search ability of the model and optimize the convergence efficiency by improving the bidirectional search strategy and applying the allocation strategy of the wolf group. In addition, we construct a comprehensive evaluation model as the evaluation standard of route optimization with the optimization goal of the shortest process time for listeners to obtain the broadcast program, the limitation of the communication media and the constraints of listening time for the listeners. To set the parameters reasonably, we apply the particle swarm optimization algorithm with a strong convergence ability to finetune the parameters. Finally, we verify the performance of the algorithm through experiments. The main contributions of our article are:

1. We propose an improved ant colony algorithm to improve the route planning of multi-media communication.
2. We adopt the phromone update strategy based on Wolf pack assignment and particle swarm to compute the optimal results while planning routes.
3. We compare with other method and achieve the best performance in the route planning.

## RELATED WORK

The general process of the traditional ant colony algorithm (*Dorigo, Birattari & Stutzle, 2006*; *Zhou, Ma & Gu, 2022*; *Deng, Xu & Zhao, 2019*) applied in route planning is as follows: initialization, searching path and updating banning table and pheromone. Firstly, we initialize the ant colony algorithm's parameters, which assumes the ants to place at the starting point. According to the rules of moving, an ant walks forward until it finds the endpoint. At the same time, we add the moving position to the banning table. The pheromone concentration and the heuristic function determine the transition probability for the ant, which selects node j from the current node i. The following formulas define the transition probability:

$$p_{ij}^{k}(t) = \begin{cases} \dfrac{[\tau_{ij}(t)]^{\alpha} \times [\omega_{ij}(t)]^{\beta}}{\sum\limits_{s \notin tabu_k} [\tau_{is}(t)]^{\alpha} \times [\omega_{ij}(t)]^{\beta}} & j \notin tabu_k \\ 0, & j \in tabu_k \end{cases} \tag{1}$$

$$\omega_{ij} = \frac{1}{dis_{ij}} \tag{2}$$

where we regard the probability of which an ant k moves from the current node i to the next node j, as $p_{ij}^{k}(t)$. Similarly, we assume the pheromone concentration of the path from node i to node j as $\tau_{ij}(t)$. $\omega_{ij}$ represents the heuristic function, whose value is the inverse of the distance $dis_{ij}$ from node $dis_{ij}$ to node j. The information heuristic factor is defined as $\alpha$, indicating the pheromone's importance. We regard the expected heuristic factor as $\beta$ to represent the importance of heuristic information on the path. Finally, $tabu_k$ is the banning list of the ant k, which represents the nodes that the ant k walks through.

To avoid too much residual information and to trap into the local optimum of the algorithm, we update the pheromone of each path after the ant cycling an iteration, whose formulas are as follows:

$$\tau_{ij}(t+1) = (1-\rho)\tau_{ij}(t) + \sum_{k=1}^{m}\Delta\tau_{ij}^{k} \tag{3}$$

$$\Delta\tau_{ij}^{k} = \begin{cases} \dfrac{Q}{L_k}, & \text{if the } k_t h \text{ ant passes } (i,j) \text{ this turn.} \\ 0, & \text{others} \end{cases} \tag{4}$$

where Q is a constant representing pheromone intensity, $\rho$ represents the pheromone volatilization coefficient whose value is between 0~1 and m refers to the number of ants. We regard the pheromone left by the ant k in the path $(i,j)$ as $\Delta\tau_{ij}^{k}$ and assume the total length of the path traversed by the ant in this cycle as $L_k$.

## AN IMPROVED ANT COLONY ALGORITHM FOR MULTI-MEDIA COMMUNICATION PATH PLANNING OF BROADCASTING AND HOSTING

In this article, we propose an improved ant colony algorithm to optimize the path planning of multi-media transmission for broadcasting and hosting, as shown in Fig. 1. The main contents of our method include three parts: the strategy of the improved two-way path search, the pheromone updating strategy based on wolves' allocation and the comprehensive evaluation model.

### An improved bidirectional path search strategy

In the traditional ant colony algorithm, all ants search from the starting point to the target. We divide all the ants equally into two groups: group A and group B. Group A searches from the starting point to the ending point. In contrast, group B searches from the ending to the starting point. When two groups of ants meet, the paths of the two groups are connected to obtain a search path, which not only improves the search efficiency but also results in the global search's insufficient ability. Therefore, we introduce the information exchange transfer factor in this article so that the two meeting ants can choose the route which has been travelled by each other or the random searching.

$$meet_{ij} = \begin{cases} 1, & \text{if } i = j \text{ or } i \text{ is near } j. \\ 0, & \text{others} \end{cases} \tag{5}$$

$$opt_{ij} = \begin{cases} 1, & meet_{ij} = 1 \text{ and } r \text{ and } > p_s. \\ 0, & meet_{ij} = 1 \text{ and } r \text{ and } \le p_s \end{cases} \tag{6}$$

where Eq. (5) represents the conditions of the two ants meeting, in which we define the $meet_{ij}$ to judge whether the ant i meets the ant j. In addition, we define the Eq. (6) to

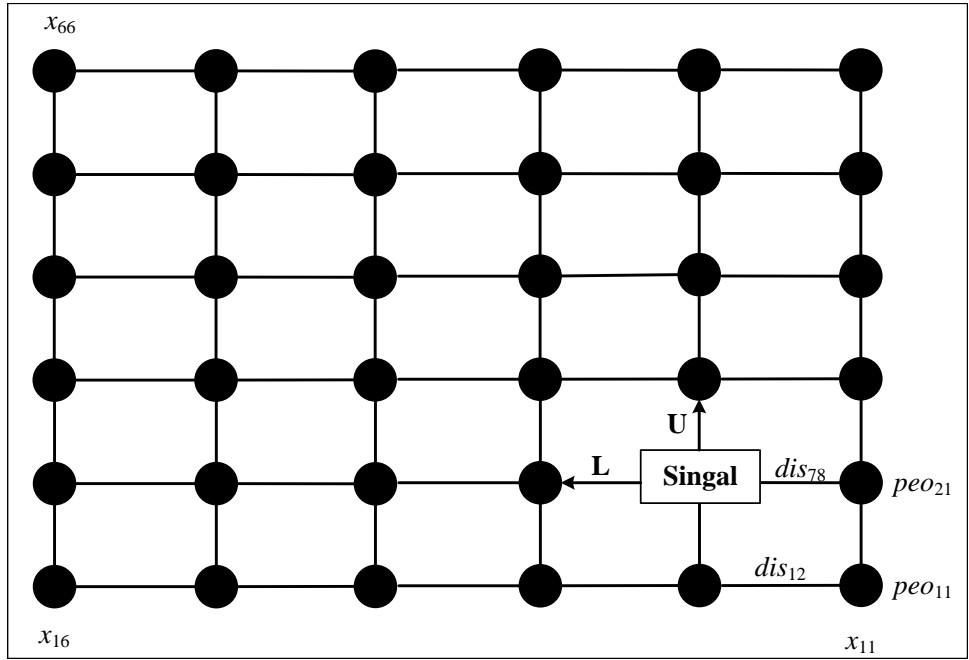

**Figure 1   The model of the path for broadcasting on multi-media.**

represent the judgment condition of whether two ants choose each other's path, of which 1 is choosing and 0 represents non-choosing. Besides, we regard the random value as rand to represent the transfer probability of information exchange. $p_s$ refers to a constant, which represents the information exchange transfer factor. If the rand is greater than this $p_s$, it means that two ants choose each other's path and transfer.

## Pheromone update strategy based on Wolf pack assignment

With the increase in the number of iterations, the pheromone of each path can be accumulated or returned to 0. We use the strategy of wolf allocation (*Zhu, Zhu & Long, 2021*; *Wang & Zhang, 2022*; *He, Tao & Luo, 2022*) to perform assignment by the updating mechanism of "survival of the strong". In the iteration process, we refer to the "reward on merit" distribution strategy in the wolves algorithm to reward the good path and punish the poor path in each iteration. After adding the Wolf pack allocation strategy, we renew the updating rules of the pheromone, which can be shown in the following formulas:

$$\tau_{ij}(t+1) = (1-\rho)\,\tau_{ij}(t) + \sum_{k=1}^{m}\Delta\tau_{ij}^{k} + \Delta^{p}\tau_{ij} + \Delta^{n}\tau_{ij} \tag{7}$$

$$\Delta^{p}\tau_{ij} = \begin{cases} \gamma\left(\dfrac{Q}{S^{p}}\right), & if\ (i,j)\ meets\ the\ best\ path. \\ 0, & others \end{cases} \tag{8}$$

$$\Delta^n \tau_{ij} = \begin{cases} \delta\left(\dfrac{Q}{S^n}\right), & if\ (i,j)\ meets\ the\ worst\ path. \\ 0, & others \end{cases} \tag{9}$$

We introduce the two parameters $\Delta^p \tau_{ij}$ and $\Delta^n \tau_{ij}$ for Eq. (7) where $\Delta^p \tau_{ij}$ is the value of the pheromone that passes through the optimal route in each iteration, as shown in Eq. (8), with the optimal route evaluation value $S^p$ of this generation. In contrast, $\Delta^n \tau_{ij}$ is the pheromone value of the worst route for $(i,j)$ in each iteration, as shown in Eq. (9), in which $S^n$ is the worst route evaluation value. And $\gamma$ and $\delta$ are constants representing the reward coefficient for the best route and the penalty coefficient for the worst route, respectively.

## Comprehensive evaluation model

We regard the heuristic function $\omega_{ij}$ as the expected degree of ant transfer from node i to node j, defined as Eq. (2) in the traditional ant colony algorithm. We consider the measuring method of a route to be not only the length of the reference route but also the number of listeners and the cost of consumption in transferring for the communication path of broadcasting and hosting programs. Therefore, we take the cost of business operating and the ratio of listeners' occupancy as the optimization objective to establish a comprehensive evaluation model for the listener number and listening time constraints. The solution of the model can reflect the comprehensive consumption transferred from node i to node j, which can be applied to improve the heuristic function.

$$eva_{ij} = a \times dis_{ij} + \frac{b}{1 + peo_j} \times dis_{ij} \tag{10}$$

$$\sum_{j=1}^{n} peo_j \leq N \tag{11}$$

$$peo_j = \sum_{i=1}^{v_j} pt_{ij} \tag{12}$$

$$\omega_{ij} = \frac{1}{eva_{ij}} \tag{13}$$

Equations (10)~(12) represent the comprehensive evaluation model and Eq. (13) represents the improved heuristic function where $eva_{ij}$ denotes the comprehensive consumption of transformation from node i to node j, $peo_j$ denotes the actual number of listeners in the next medium j, N denotes the expected number of broadcasting host programs, n is the total number of media, $V_j$ indicates the expected audience of the medium j and $pt_{ji}$ indicates the listening time of the i-th person in the medium j. Besides, $pt \in [T, T+X]$ where T represents the start time of the current broadcast and X is the uncertain end time of the broadcast. In the ending, a and b is the adjustment parameter whose value is from 0 to 1. When the value of a is 1 and the value of b is 0, Eq. (13) degenerates into Eq. (2).

## Parameter optimization based on improved particle swarm

The parameter value in the ant colony algorithm has a tremendous impact. Therefore, we employ the particle swarm's strong global convergence ability to optimize the important parameters in the ant colony algorithm (*Mousa & Hussein, 2022*; *Shami, El-Saleh & Alswaitti, 2022*). Furthermore, we can complete the parameter optimization of the algorithm by repeatedly calling the ant colony algorithm in the process of multiple iterations. In this process, Eq. (15) is used as the standard to judge the quality of the parameters.

$$syn(M_t) = \sum_{i,j \in M_t} eva_{ij} \tag{14}$$

$$fit = \lambda \times \frac{1}{syn(M_t)} + \mu \times ite. \tag{15}$$

In Eq. (14), we set the $i, j$ to be the connected node for the path $M_t$ and define the $eva_{ij}$ in Eq. (10). In Eq. (15), fit represents the fitness function value, $syn(M_t)$ is calculated from Eq. (14) and ite is the minimum number of iterations when the algorithm obtains the optimal solution. In addition, $\lambda$ and $\mu$ are constants with values from 0 to 1, representing the weight coefficient of the fitness function and the minimum number of iterations.

Firstly, we initial the particle populations. Subsequently, we bring the parameters searched by each particle to the improved ant colony algorithm. Finally, we compute the fitness of particles and select the optimal parameters following Eq. (15). During the search process, the moving speed and position of each particle are varied through the following formula:

$$v_i(k+1) = \varepsilon v_i(k) + c_1 r_1 \left[ pbest_i(k) - x_i(k) \right] + c_2 r_2 \left[ gbest_i(k) - x_i(k) \right] \tag{16}$$

$$x_i(k+1) = x_i(k) + v_i(k+1) \tag{17}$$

where $c_1$ and $c_2$ denotes learning factors and $\varepsilon$ represents inertia factor. In addition, $r_1$ and $r_2$ The random numbers ranging from 0 to 1 are used to adjust the search space of particles. We regard the individual and group optimal values as $pbest_i$ and $gbest_i(k)$, respectively.

## RESULTS AND ANALYSIS

To verify the performance of the improved algorithm, we use Matlab simulation to conduct experiments. Firstly, multiple road network models with different scales are constructed. Then, the advantages of the improved algorithm are evaluated from two aspects: shortest path planning and optimal comprehensive evaluation path planning.

### The shortest path planning

We consider that the communication path planning with the new media medium can be influenced by the propagation path length, listening to the number of factors and so on for the broadcasting programs. Therefore, we regard the propagation path length as the only

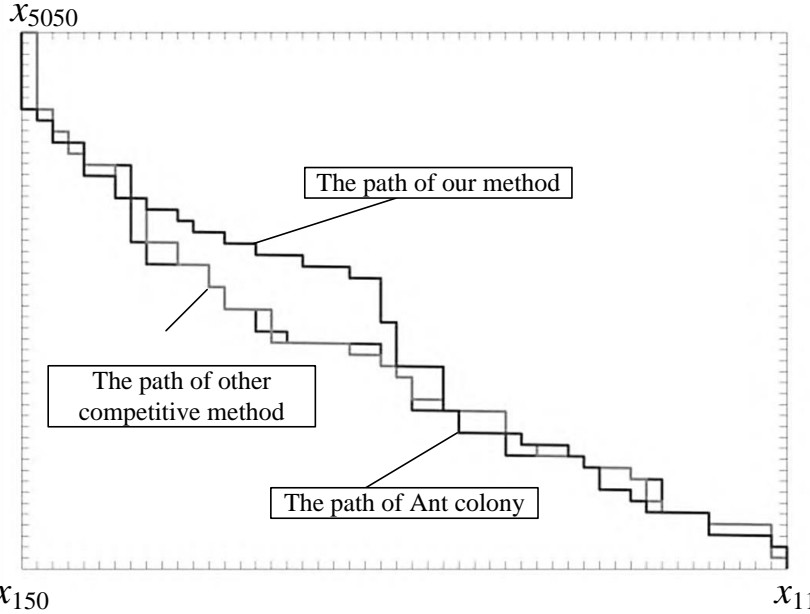

**Figure 2 The optimal path of three algorithms for broadcasting on multi-media.**

evaluation metric to validate the simulation. To reflect the performance of the improved algorithm, we select the structure of the route network with a scale of $50 \times 50$ for simulation.

Firstly, we present the path results obtained by the traditional ant colony algorithm, *Yunlong & Ying (2009)* with bidirectional search strategy and our proposed algorithm in Fig. 2. At the same time, we show the optimal path length and the number of iterations in Fig. 3. We can demonstrate that the global search ability of the proposed algorithm is significantly improved, by which we can obtain the shorter path. However, the improvement of convergence ability is insufficient.

The traditional ant colony algorithm can achieve the optimal path length of 302.4. The path length corresponding to *Yunlong & Ying (2009)* is 290.94. In contrast, *Yunlong & Ying (2009)* improved the global search ability. Our proposed algorithm further enhances the global search ability and achieves the optimal path with a length of 285.88.

In addition, the traditional ant colony algorithm converges after 41 iterations, the algorithm used in *Yunlong & Ying (2009)* converges after 29 iterations and the algorithm that we propose converges after 36 iterations. Analysis shows that *Yunlong & Ying (2009)* and the proposed algorithm have improved the convergence speed compared with the traditional algorithm. However, the convergence speed of the algorithm that we propose is slightly less than *Yunlong & Ying (2009)*.

## The path planning of the comprehensive model

According to the conclusions above, the proposed algorithm has significantly improved compared to the traditional ant colony algorithm. In the transmission path problem of broadcasting and hosting programs, the path length is no longer the only indicator, but

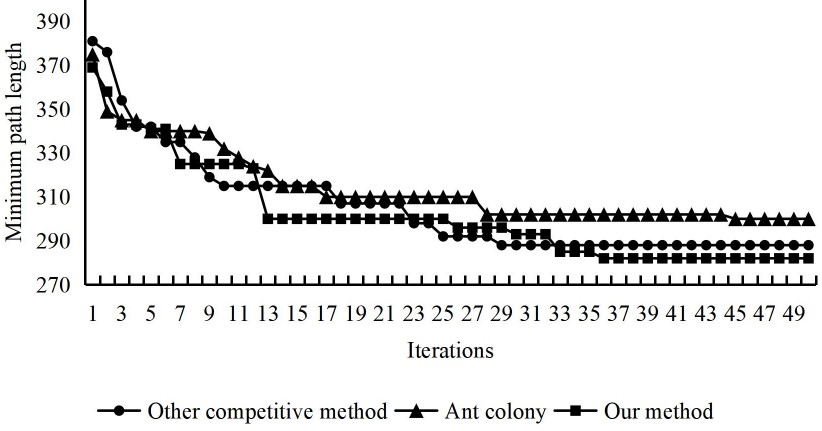

**Figure 3** Comparison with other methods.

**Table 1** Comparison with Ant colony in terms of metrics.

| Methods | Length of path | Number of listeners | Comprehensive score | Cost time | Iterations |
|---|---|---|---|---|---|
| Ant colony | 42.3 | 16 | 8.7 | 0.439 | 6 |
| Ours | 45.6 | 19 | 7.2 | 0.355 | 4 |

also the ratio of the listening should be considered. We should only reach the balance point between them to achieve the maximum profit.

To verify the superiority of the comprehensive performance for our proposed algorithm, we apply the comprehensive evaluation function as the fitness function's evaluation metric for verification simulation. As shown in Eq. (14), the lower the overall score is, the better the path can be. To prove the algorithm's effectiveness more intuitively, we choose a simple route network structure of 6×6.

Firstly, we present the path sought by the traditional ant colony algorithm and our proposed algorithm in Fig. 4. In addition, we list the various indicators of these two algorithms in Table 1. We can see from Table 1 that the path obtained by the algorithm that we propose is slightly longer than the traditional algorithm in length. But the number of listeners is significantly increased and the comprehensive score is lower than the traditional algorithm. Therefore, we can demonstrate that the algorithm in this article, which takes the comprehensive score as the standard, achieves great progress with the propagation distance and the number of listeners. At the same time, we boost the comprehensive optimization performance while comparing it with the traditional algorithm.

With the analysis, we can find that the 6×6 route network structure is relatively simple. These two algorithms can both traverse all paths. The traditional algorithm uses propagation distance as the only metric to obtain the shortest path of the model. However, we can achieve the optimal path of the route network model with the two indicators after considering the propagation distance and the number of listeners. With this comparison,

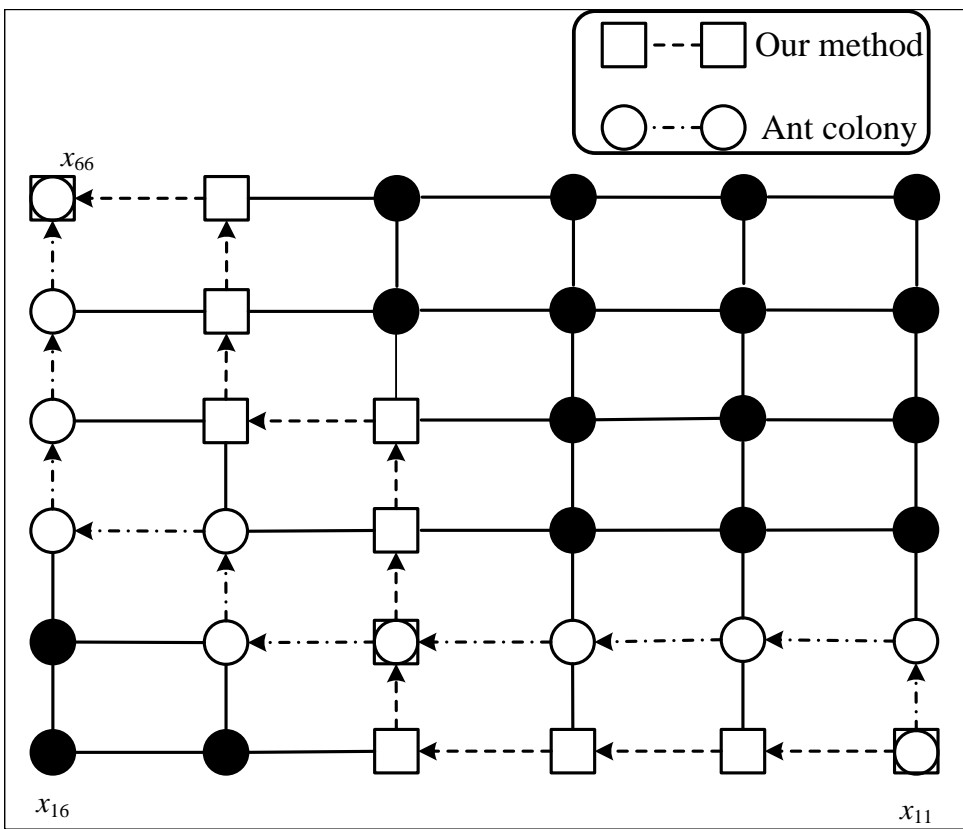

**Figure 4  Paths solved by ant colony and our method.**

the algorithm in this article is more satisfied with the company's interests and achieves the unity of rapid dissemination and economic benefits.

## CONCLUSION

In this article, we research the ant colony algorithm for the transmission path planning problem of broadcast and host programs with the new media while suffering from public health events in a dynamic environment. Firstly, we introduce an improved bidirectional search strategy to expand the search range early in the algorithm. Then, we apply the allocation strategy of the wolf group to improve the pheromone update principle and to enhance the global search ability further. In addition, we establish a comprehensive evaluation model which not only reduces the operating cost but also increases the listening rate of the program. Finally, we employ the improved particle swarm algorithm for optimization and bring the comprehensive evaluation metric into the fitness function to adjust the optimal parameters. We can demonstrate that the improved algorithm significantly improves the shortest path planning and comprehensive evaluation metric path planning. However, the convergence speed of the algorithm should be strengthened. In the future, we will explore extending the broadcast and host programs to all types of programs and continue to research the attention to find more important points in the

path planning of broadcast and host programs with the new media (*Hölscher, Tenbrink & Wiener, 2011*; *Zheng et al., 2005*; *Lo, Chen & Hu, 2019*).

### Funding
The author received no funding for this work.

### Competing Interests
The author declares that there are no competing interests.

### Author Contributions
- Qiong Hu conceived and designed the experiments, performed the experiments, analyzed the data, performed the computation work, prepared figures and/or tables, authored or reviewed drafts of the article, and approved the final draft.

### Data Availability
The code is available in the Supplemental Files.

The data is available at Kaggle and Zenodo: Available at https://www.kaggle.com/datasets/mehdimka/path-planning

None. (2023). Dataset [Data set]. Zenodo. https://doi.org/10.5281/zenodo.7818515.

### Supplemental Information
Supplemental information for this article can be found online at http://dx.doi.org/10.7717/peerj-cs.1397#supplemental-information.

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
