# Peer review of "Sustainable development of multi-media communication path of broadcasting and hosting with a dynamic environment"

_PeerJ Computer Science, doi:10.7717/peerj-cs.1397_

## Round 0.1 · original submission · Major Revisions

Dear colleagues,

Your paper needs major improvements as suggested by the experts. The language of the paper is also weak, so please get it professionally proofread.

Reviewer 1 ·

Basic reporting

An improved integrated path planning algorithm based on ant colony algorithm is proposed to find the optimal path of multimedia transmission for broadcasting and hosting programs in dynamic environment. Firstly, the bidirectional search strategy is improved, probabilistic transition is optimized, early search scope is expanded, and global search capability is enhanced. Then, they use the Wolf's allocation strategy to change the pheromone update rules to enhance the convergence performance of this
method. Finally, author construct a comprehensive evaluation model with the objective of
minimizing the process time for listeners to acquire broadcast programs.
The academic English is very weak, extensive English revision is required, proofreading is required
In the academic English we prefer using the passive voice( third person), trying to avoid (we, they, she, he,....etc)

Experimental design

This is a valuable and interesting idea. However, with the current quality, this article cannot be published.
The experiments design could be explained using pseudocodes or flowchart to make it easy for the reader to understand the module.

Validity of the findings

More details are required for the second scenario.

Additional comments

This article has many defects, so my suggestion is a minor revision.
1. The abstract must be self-contained and concisely describe the reason for the work,
methodology, results, and conclusions;
2. The content of the abstract should be reduced to 200 words;
3. The research introduction of ant colony algorithm in the citation should be placed
in Section 2.
4. The writer should check the statements in the paper carefully. For example, "And
now, it is mainly applied to the problems such as vehicle transportation scheduling and
path planning. "This statement has little relevance to the research content of the
subsequent content.
5. What does "other competitive method" in Figure 3 specifically refer to?
6. Does Figure 2 and Figure 3 use the same method for comparison? If so, they need
to correspond to each other and check them further;
7. Add more models for comparison in Table 1, such as ABC algorithm;
8. In summary, the "dynamic environment" in the title of the paper is not obvious in
the manuscript, so it is suggested to modify the title

Reviewer 2 ·

Basic reporting

In this study, the author studied the ant colony algorithm to solve the transmission path planning problem of new media broadcasting and hosting programs.
At the same time, the advantages and disadvantages of traditional ant colony algorithm are analyzed, and an improved ant colony algorithm is proposed.

This manuscript may be published in this journal after addressing the following comments;

1. The description of experimental results in the abstract is too thin, so some descriptive data should be added;
2. Highlight the definition and main innovation of the problem studied;
3. Please sort out the article again and summarize 4-5 representative keywords
4. Some descriptions need to be deleted in close connection with the current international situation.“People have been spending more time at home in recent years due to public health events, such as the coronavirus pandemic”;
5. The introduction of the introduction part coincides with the description of the literature review in the second chapter.
6. It is suggested that the author sort out the citations of these two parts again to ensure a certain logic of writing;
7. What has been done in the past research on the optimization of ant colony algorithm? They need to be reasonably summed up;
8. Figure 1 requires more description to specifically clarify the main design ideas of the model;
9. It is suggested to use different colors in Figure 2 to distinguish the optimal path of the three multimedia playback algorithms;
10. There are still some language expression problems in this article, which need to be further modified. Please check the full text and deal with it.

Experimental design

The experimental results are well explained. The experimental results show that the improved algorithm has remarkable improvement in the shortest path planning and the comprehensive evaluation and measurement path planning. But the convergence speed of the algorithm needs to be strengthened. Figure 1 requires more description to specifically clarify the main design ideas of the model. It is suggested to use different colors in Figure 2 to distinguish the optimal path of the three multimedia playback algorithms.

Validity of the findings

The results among the study participants represent true findings.

---

## Round 0.2 · accepted · Accept

Thanks for incorporating the experts comments, I'm happy to accept your article.

Reviewer 1 ·

Basic reporting

All the required editing has been done based on the commented provided in the first round, so I am happy to accept the paper

Experimental design

All the required editing has been done based on the commented provided in the first round, so I am happy to accept the paper

Validity of the findings

All the required editing has been done based on the commented provided in the first round, so I am happy to accept the paper

Additional comments

All the required editing has been done based on the commented provided in the first round, so I am happy to accept the paper

Reviewer 2 ·

Basic reporting

All the comments are incorporated successfully
and article is ready for publish.

Experimental design

All the comments are incorporated successfully
and article is ready for publish.

Validity of the findings

All the comments are incorporated successfully
and article is ready for publish.

Additional comments

In this article the author proposed a model for broadcasting and hosting of transmissions in dynamic environment by using evolutionary algorithm "Ant Colony Algorithm"
This is an optimal approach that optimizes the routes. This is such a great contribution because evolutionary algorithms are the best algorithms that solve the np hard abd np complete problems.